# Current Status of Brain Tumor in the Kingdom of Saudi Arabia and Application of Nanobiotechnology for Its Treatment: A Comprehensive Review

**DOI:** 10.3390/life11050421

**Published:** 2021-05-05

**Authors:** Afrasim Moin, Syed Mohd Danish Rizvi, Talib Hussain, D. V. Gowda, Gehad M. Subaiea, Mustafa M. A. Elsayed, Mukhtar Ansari, Abulrahman Sattam Alanazi, Hemant Yadav

**Affiliations:** 1Department of Pharmaceutics, College of Pharmacy, University of Hail, Hail 81442, Saudi Arabia; a.moinuddin@uoh.edu.sa (A.M.); mu.elsayed@uoh.edu.sa (M.M.A.E.); 2Department of Pharmacology and Toxicology, College of Pharmacy, University of Hail, Hail 81442, Saudi Arabia; g.subaiea@uoh.edu.sa; 3Department of Pharmaceutics, JSS College of Pharmacy, Mysuru 570015, India; dvgowda@jssuni.edu.in; 4Department of Clinical Pharmacy, College of Pharmacy, University of Hail, Hail 81442, Saudi Arabia; m.ansari@uoh.edu.sa (M.A.); ar.alanazi@uoh.edu.sa (A.S.A.); 5Department of Pharmaceutics, RAK College of Pharmaceutical Sciences, RAK Medical & Health Sciences University, Ras Al Khaimah 11172, United Arab Emirates; hemant@rakmhsu.ac.ae

**Keywords:** brain cancer, gliomas, nanobiotechnology, polymeric nanomaterials, Saudi Arabia

## Abstract

Objective: Brain tumors are the most challenging of all tumors and accounts for about 3% of all cancer allied deaths. The aim of the present review is to examine the brain tumor prevalence and treatment modalities available in the Kingdom of Saudi Arabia. It also provides a comprehensive analysis of the application of various nanotechnology-based products for brain cancer treatments along with their prospective future advancements. Methods: A literature review was performed to identify and summarize the current status of brain cancer in Saudi Arabia and the scope of nanobiotechnology in its treatment. Results: Depending upon the study population data analysis, gliomas, astrocytoma, meningioma, and metastatic cancer have a higher incidence rate in Saudi Arabia than in other countries, and are mostly treated in accordance with conventional treatment modalities for brain cancer. Due to the poor prognosis of cancer, it has an average survival rate of 2 years. Conventional therapy includes surgery, radiotherapy, chemotherapy, and a combination thereof, but these do not control the disease’s recurrence. Among the various nanomaterials discussed, liposomes and polymeric nanoformulations have demonstrated encouraging outcomes for facilitated brain cancer treatment. Conclusions: Nanomaterials possess the capacity to overcome the shortcomings of conventional therapies. Polymer-based nanomaterials have shown encouraging outcomes against brain cancer when amalgamated with other nano-based therapies. Nonetheless, nanomaterials could be devised that possess minimal toxicity towards normal cells or that specifically target tumor cells. In addition, rigorous clinical investigations are warranted to prepare them as an efficient and safe modality for brain cancer therapy.

## 1. Introduction

Even though there have been advances in the treatment of cancer, brain cancer remains a persistent challenge due to its complicated, heterogeneous, and aggressive nature. According to the American Cancer Society, by 2040, cancer cases would rise to quite greater extent and will record about 27.5 million [1,2]. Three percent of total cancer deaths reported are related to the central nervous system, with the death rate being more prevalent among men [3,4]. Brain cancer ranks are considered the most elevated among pediatric cancer-related deaths [5]. The WHO broadly classifies brain tumors based on the area affected, such as the spine and cranial nerves, neuroepithelial tissue, hematopoietic neoplasms, and lymphomas. The few non-classified tumors include meningioma, glioblastoma and glioma, pituitary and craniopharyngioma tumors, nerve sheath tumors, heterotopias, neoplasm, germ cell tumors, and hemangioma [6]. Glioblastoma is the most aggressive of its form, and affects two-thirds of the total adult brain cancer population with an average survival rate of 2 years following diagnosis. According to the WHO, gliomas can be classified as grade I to grade IV, i.e., pilocytic, diffuse and anaplastic astrocytoma, and glioblastoma multiforme, with the latter being a high-grade malignant glioma with poor prognosis. The metastatic primary brain tumor glioblastoma multiforme (WHO Grade IV) is responsible for about 12–15% of total brain tumors, and it is the most prevalent cancer of the brain [7]. Brain metastasization can occur from prostate, breast, colorectal, renal or melanoma, and lung cancer; moreover, breast cancer is responsible for the majority (about 70%) of brain metastases [8,9]. The existing therapy for glioma following surgical removal consists of radiochemotherapy. Temozolomide adjuvant chemotherapy is a customized model that has progressed through phase III trials that demonstrates a 27.2% survival rate, compared to glioblastoma patients treated with postsurgical radiotherapy, who exhibit only a 10.9% survival rate, as evaluated by the National Cancer Institute of Canada (a European Organization for Research and Treatment of Cancer) in 2004 [7,10]. There is no specific treatment regimen for the condition’s recurrence, and experimental agents that are still in clinical trials are widely used to treat patients [7,11]. Oligodendroglioma, which results from oligodendrocytes, produces myelin and accounts for 9% of tumors in adults and about 4% of primary CNS tumors in infants. Ependymomas account for about 8–10% of CNS cancers in children and develop from the brain’s ependymal cells, ventricles, and spinal cord, consisting of cerebrospinal fluid (CSF) [12].

Children with brain cancer are often affected by the post-treatment side effects of surgery, chemotherapy, and radiotherapy [13,14]. Due to their biological characteristics, therapy for brain tumors is quite tough, and the tumors’ infiltration into vital organs makes them more resilient to treatment. The blood–brain barrier (BBB) protects complex tumors beneath it, protecting sensitive neural tissues from exposure to external influence and accessibility by drugs [15]. The barrier limits the effect of therapeutic agents systemically applied in the brain tumor cell barrier (BTB) of the brain as a result of the effluent activity of tumor cells. Dose-restrictive and primarily myelosuppressive toxicity, as well as tumor tolerance towards alkylating compounds, mainly via the nucleotide repair enzyme MGMT (O6-methylguanine-DNA methyltransferase), are further problems associated with the successful management of brain tumors [16,17]. The brain’s exclusive genetic, developmental, microenvironmental and epigenetic characteristics often make these tumors resilient to traditional and innovative treatments [18,19]. Due to their location and susceptibility to undergoing malignant transformations over time, a benign neoplasm in the brain may have catastrophic effects and become lethal. Furthermore, the prevalence of brain tumors is increasing globally, primarily as a result of modern medical methods and imaging methods [20]. The development of effective chemotherapy for brain tumors is lagging behind all other cancers as well as systemic chemotherapy for glioblastoma. The most aggressive form of cancer therapy that has been proven to be effective is temozolomide. However, its beneficial effects are only visible after months of treatment [21,22].

Several vital advances in therapeutic methods have occurred in clinical oncology during the last decade, based on a patient’s genetic makeup and biology, as well as targeted therapy [23]. In cancer treatment, identifying cancer as early as possible is of paramount importance, because it is much easier to treat cancer at a very early stage. In particular, cancer cells have a reduced risk of having mutations at that stage, which accounts for drug resistance development [24]. Although various efforts to understand the microenvironment and physiology of brain cancers are underway, facilitating the identification of cancer-specific regulators subjected to multiple pharmacological interventions is one of them, where the limiting factor is the lack of an efficient drug delivery system [25,26]. Conventional diagnostic and therapeutic methodologies like MRI (Magnetic Resonance Imaging), radiotherapy, and surgical resection have demonstrated several limitations in clinical trials allied to their imaging sensitivity and therapeutic efficacy. Developments of advanced imaging and diagnostic approaches, for instance, functional MRI, positron emission tomography (PET) and computed tomography (CT) have made an excellent contribution in tumor detection and improved treatment [27,28,29,30,31]. Surgical removal can effectually alleviate brain pressure and symptoms, and lead to incomplete clearance of the lesions and tissue damage with localized bleeding. On the other hand, radiation therapy is ideal for tumors of varying sizes that cannot undergo surgical resection [32,33]. However, it can simultaneously destroy the normal tissues leading to severe adverse effects on the onset of care. Apart from that chemotherapeutic agents also cause metabolic disorders, inhibition of cell cycle, and systemic toxicity. Thus, different therapeutic and investigational approaches adopted for brain cancer therapy are seemed to be ineffective for several reasons, and have not met the anticipated safety and efficacy [34].

Nanomedicine uses versatile, spatially organized and engineered complex nanomaterials to attain diverse therapeutic benefits at the molecular level [26]. Contrast agents are used as the diagnostic tool for cancer-specific investigations and therapy and hold different drug-delivery applications [35]. With the BBB crossing ability and targeted delivery of the drug, nanoparticles remain promising to transport and effectually deliver medications to the brain cells. Nanomaterials have received much attention in developing nanotherapeutics for gliomas in the past few years [36]. For oncologists, the delivery of targeted cancer therapeutics is often considered as a problem; however, nano vehicles include a matrix or a core with the iatric molecule, sometimes even with the surface modifications. Nano delivery systems have many benefits over traditional treatments, such as preventing drug degradation inside the body, increasing drug absorption, and inhibiting undesirable drug interaction with normal cells. The most favored sizes of nanoparticulate drug delivery systems are 40–100 nm, 80–200 nm, 20–60 nm, for polymeric nanoparticles, liposomes and micelles, respectively; and the smallest amidst them are dendrimers being <10 nm in diameter. The provision of anticancer therapeutics is endorsed by numerous studies, which are also being tested for clinical trials [37,38].

The present review delivers a broader outline of brain cancer and its current status in Saudi Arabia. It offers a meta-analysis of its prevalence, adopted treatment methods, and the need for alternative strategies for brain cancer therapy. It also broadly comprehends the potential benefits allied with and incorporation of nanobiotechnology for augmented and more circumspect brain cancer therapy.

## 2. Materials and Methods

A systematic review was carried on the current status of brain cancer in Saudi Arabia. In the present review, efforts have been made to cover topics pertinent to brain cancer and its treatment modalities spanning for last two decades. The in-depth literature mining was done from scientific databases available in Saudi digital library such as Web of Science, Scopus, PubMed, etc. The terminologies used for data mining were brain cancer, gliomas, nanobiotechnology, polymeric nanomaterials, Saudi Arabia, liposomes, etc., to list a few. The manuscript dealt in detail on glioma with a comprehensive analysis of the application of various nanotechnology-based products for facilitating the treatment and allied prospective advancements for implications in the Kingdom of Saudi Arabia. An ample of full-text articles were considered and critically analyzed in the present study. The consistency in the significant findings of these studies was critically evaluated and conclusions drawn are presented in this review.

## 3. Brain Cancer in Saudi Arabia

### 3.1. Incidences of Brain Cancer in Saudi Arabia

The global age-standardized (ASR) incidence of brain and neurological cancer in extremely high/high HDI (Human Development Index) areas vs. medium/low HDI areas was reported to be 2.4 and 5.0 for males and 1.7 and 4.0 for females, respectively. It is noteworthy that under the 4-tier structure of the United Nations Development Program, Saudi Arabia has been ranked as having very high HDI. The incidence rate is noted to be quite high in HDI countries on par with the low, medium HDI countries, where males have a higher incidence than females [39,40]. From 1998 to 2007, brain tumors were the 10th most prevalent tumors in the Gulf Cooperation Council (GCC) countries, and the average age-standardized prevalence (ASR) for brain cancer was found to be 2.4 and 1.6 for males and females per 100,000 population, respectively [22,40,41]. The largest nation in the Middle East, with an approximate population of around 22 million, is Saudi Arabia. According to the recent Saudi Cancer Registry study, executed in 2014, about 329 new brain cancer cases accounted for 2.8% of total cancer patients. The Riyadh Region, followed by the Jouf Region, the Northern Region, the Eastern Region, and the Qassim Region, were the five regions with the highest ASR for brain cancer. Whereas after the Jazan, Hail, and Asir areas, the Madina region had the fewest ASR for brain cancer [22].

A ten-year span study was conducted at the National Neuroscience Institute (NNI) of King Fahad Medical City, comprising 992 cases of primary cancers affecting the CNS, and accounting for 278 (28.02%) and 714 (71.97%) of pediatric and adults, respectively. It has been noted that non-malignant and malignant tumors had affected about 60.08% of the adult and pediatric population, and about 61.37% had been dominated. The incidence of CNS tumors such as meningioma and glioblastoma have been mentioned to account for 26.99% and 25.10%, respectively. The study outcomes have also revealed that glioblastoma among the adult population (54.52%) was the most common of all the adult glioma (46.49%) that affected the population. In contrast, among the pediatric population, medulloblastomas (26.62%) had a higher incidence, followed by pilocytic astrocytoma (17.99%) and ependymomas (11.51%) with high-grade glioma (10.07%) [40]. In a 2019 retrospective study conducted at King Fahad Hospital, Madinah, by Mohammed et al., the brain tumor cases’ epidemiology for 12 years from January 2006 to December 2017 were studied. Among 227 reported cases, 53.7% were males, and 46.3% were female patients with brain cancers, of which the population composed of 10.6% of pediatric and 89.4% comprised of the adults. Histopathological findings have shown that the most prevalent age group affected was between 40 and 49 years, comprising 23.5% of the brain tumor population. Histopathologic examinations have depicted that 30.8% were affected with meningioma among the total cancer patient, followed by the astrocytic (majority of the cases were WHO grade IV), metastatic and embryonal tumors accounting 29.1%, 7.7%, and 6.6%, respectively. Among which the meningothelial meningioma (48.5%) was most common among meningioma [22].

The King Abdulaziz University Hospital (KAUH) in Jeddah, Saudi Arabia, conducted an 8-year analysis of 112 brain cancer patients, including 77 adults and 35 children. The overall prevalence rate and the average annual incidence were 54.8 × 10^5^ and 6.0 × 10^5^ for the tumor population. Notably, the outcomes have revealed that the average yearly brain tumor incidences were 3.7 × 10^5^ and 2.4 × 10^5^ among the male and female populations. The prevalence of brain tumors in adults was 59 × 10^5^ at KAUH. The prevalence rate for adult males was noted to be 32.9 × 10^5^, and the prevalence rate for adult females was noted to be 26.1 × 10^5^. The adult population’s average annual prevalence rate was 6.5 × 10^5^. The estimated annual incidence rate was 4.1 × 10^5^ for adult males, and the average annual incidence rate was 2.4 × 10^5^ for adult females. The most prevalent brain tumor forms noticed in the adult population were owing to metastasis, contributing to about 28.57%. Jeddah (with 80.6%) was the most frequent residence place for patients with brain tumors at KAUH, followed by Makkah (with 14.4%). Saudis (36.36%) was the most affected nationality found in the results, followed by Yemeni (28.57%). It has been reported that the metastatic brain tumor was the most prevalent tumor affecting adult Saudis at KAUH (25%), followed by astrocytoma and meningioma at 17.8% each. Astrocytoma (25%) was the most common tumor among adult Yemenis at KAUH, followed by pituitary adenoma and metastatic brain tumors at 14.2% incidence rate for each. In KAUH, the pediatric brain tumor prevailing incidence was 47 × 10^5^. The prevalence of male pediatric patients was about 27 × 10^5^, and the majority was 20 × 10^5^ for females. Pediatrics has an average annual occurrence of 5.2 × 10^5^. Pediatric males had an average occurrence rate of 3.0 × 10^5^ per annum, while the average pediatric females had it about 2.2 × 10^5^ per annum. KAUH’s most prominent pediatric brain tumor was astrocytoma with recorded incidence rate of 37.1%. Most patients suffering from childhood brain tumors were from Jeddah (85.71%) followed by Makkah (11.43%) [42]. Of the total 149 cases, there were 96 adults and 53 pediatrics, accounting for 58% of the male and 42% of the female populations. The adult and pediatric cases have 43.7% and 32% glioblastoma multiform (WHO IV), 6.2% and 12.3% of primitive neuroectodermal tumor (PNET), 9.3% of pilocytic astrocytoma, 9.3% and 8.7% of anaplastic astrocytoma, respectively [43].

### 3.2. Treatment Strategy Followed for Brain Cancer in Saudi Arabia

According to a retrospective study performed at Extensive Cancer Center in King Fahad Medical City in Riyadh (Saudi Arabia), approximately 81% of Glioblastoma Multiforme (GBM) patients received surgical resection, while the remaining 98% underwent radiotherapy from 2008 to 2013. About 72% of the patients underwent radiotherapy with TMZ supplementation. It has been reported that the patients had an average survival period of 13.7 months, and that the combined modality treatment accounted for about 19.7 months of better results without persistent illness with TMZ adjuvant therapy [44]. In 1994, King Faisal Specialist Hospital and Research Center in Riyadh (Saudi Arabia) has conducted a study on 34 patients with primary brain tumors. The findings revealed that approximately 23 patients had received a ventriculoperitoneal shunt before any treatment. Patients with brain cancer had received radiation therapy as an adjuvant procedure (about 22 patients), including chemotherapy in 3 cases. On follow-up for 3 to 5- months, the mortality rate was noted to be 5.9% over a period of two months [45]. Till now, conventional therapies have been implemented to treat brain cancer patients in Saudi Arabia.

### 3.3. Success and Failure Rate of Brain Cancer Treatment

A multidisciplinary approach is required to treat newly diagnosed GBM. The current routine care consists of full safe resection procedure accompanied by concomitant temozolomide (TMZ) adjuvant chemotherapy along with the radiation. Extensive and complete operative GBM resection is problematic since these cancers are typically invasive and found in eloquent brain areas such as spoken, motor-functioning, and sensory control areas. The primary cancer resections are not curative because of the high intrusiveness, and the infiltrating cancer cells still reside inside the surrounding brain tissues resulting later in recurrent diseases [46]. Negative findings have been associated with tumors that are > 5–6 cm in size, and those exceeding the midline [47]. Depending on the location, grade, and morphology of cancer, surgical cancer resection could be determined. Patients suffering from high-grade cancers need almost the entire surgical removal of the tumor, which increases the survival rate to reduce the cancer burden and pressure within the skull [48,49]. Multiple trials have indicated the value of aggressive surgical resection, where it is possible to extend patient’s survival rate, with improved degree of resection. A statistically meaningful correlation between a higher degree of surgical removal and overall survival rate or progression-free survival (PFS) have been established in many studies [50,51]. Innovations in pre-operative detection and surgical methods have attained complete resection while retaining health and life quality [52]. However, the surgical resection cannot treat brain cancer completely because of its invasive nature, and relapsation is also observed in approximately 80% of cases within 2 or 3 cm of the original tumor margin [53].

Different modes of neuronavigation and incorporation of patient-centric functional and anatomical data have been made possible by the application of diffusion tensor imaging and functional MRI during pre-operative preparation along with the CT scans, MRI and ultrasound with direct stimulation throughout the surgery. Despite this, distinguishing the normal cells and cancer cells is still challenging, and fluorescence guidance using 5-aminolevulinic acid dye (5-ALA) was more successful than traditional neuronavigation-guided surgery [54]. The cost of advanced instruments, operators, and surgical suites also includes the drawbacks of these new technologies. Further trials are essential for clarifying clinical advantages before a standard of treatment is developed for all brain cancer patients. Stereotactic Radiosurgery (SRS) can treat recurrent GBM with external beam radiation therapy (RT) replacement. Nevertheless, SRS’s use to treat recently diagnosed cancerous gliomas has been under consideration [55]. Some limitations are associated with RT, such as damage of neurons, normal brain cell necrosis, and resistance of tumor cells towards radiation [56].

During 25 to 35 daily treatments for (5-7) weeks, regular treatment with outside RT, the total dose to be administered was regulated by several factors: the location of the tumor, grade, histology, and resection rates [55]. Even with surgical removal advancements, the prognosis for glioma patients stays low, with about 15-month survival rates. Supratentorial (cerebral) and cerebellar cancers, which are more appropriate for surgical care, have a more robust prognosis than brain or diencephalon cancers. According to Johnson and O’Neill (2012), a development in overall survival rate was exhibited after continuous multi-modality therapy [57]. Until 2005, standard treatment findings of a pivotal phase III analysis GBM treatment was postoperative RT alone. However, one research study has demonstrated the efficacy of external RT beam with TMZ chemotherapy than RT alone [21].

### 3.4. Need for Alternate Treatment Strategy?

The latest findings in GBM therapy are inadequate. The overall survival rate for patients suffering from primary GBM is 5 months, and the secondary GBMs is eight months; only about 2% of the total patients reach 5 years of survival. The advent of the temozolomide (TMZ) as an alkylating agent, along with concurrent radiotherapy accompanied by monotherapy in a period of 28 days, extends individual patients’ lives (Stupp protocol-Stupp regimen), has achieved the most significant breakthrough in GBM therapy in the last 15 yrs. However, this treatment approach does not offer the patient a more prolonged period free of diseases, as recurrence happens on an average within 7 months of patients that reacted well to this therapy [58,59]. Two methods *viz*. BBB crossing and evading the BBB can be adopted for delivering drugs to the brain. In fact, 5 major mechanisms can be used to cross the BBB: adsorptive transcytosis (AMT), paracellular transport, carrier-mediated transport (CMT), receptor-mediated transcytosis (RMT), and passive transcellular diffusion (Figure 1). The natural BBB physiology does not give access to paracellular transportation permeability [60]. Moreover, it could occur due to the damage of BBB in CNS conditions like GBM, allowing brain drug delivery [61], as the BBB is composed of tight junctions and adherent junctions mostly made of endothelial cells restricting the permeability of the erratic molecule towards brain. As the brain cancer cells were surrounded by various cells like microglia, astrocytes and pericytes, which makes them more specialized, inducing the blood vessels endothelium of the brain. Additionally, expression of various proteins on the surface of BBB has been observed, like, multidrug resistance proteins 4 (MRP4), ATP-binding cassette (ABC) efflux pumps, P-glycoprotein (P-gp), and breast cancer resistance protein (BCRP). Unfortunately, about 90% of the therapeutic drugs form substrates for the efflux pumps and cannot reach the tumor cells [62]. Owing to brain cancers’ invasiveness and complex nature and the resistance developed against chemotherapy and radiation therapy, brain tumor treatments are rendered ineffective. Hence, there is a need for an alternative strategy for brain cancer treatment.

## 4. Scope and Application of Nanobiotechnology in Brain Cancer Treatment

### 4.1. Technological Advancements

The advancement of technology has led to cancer diagnosis and treatment through approaches at the nanoscale, which has successfully established various bioimaging and treatment modalities. The most favorable characteristics exhibited by the nanomaterials are large surface area to volume ratio, small size, distinct characteristics, and a quickly modifiable surface with different molecules, making it an excellent transport vehicle that overcomes the various demerits of the conventional treatment methods which can bypass the BBB, skin–tissue barriers. As it can overcome the biggest challenges of brain cancer treatments, it provides an excellent visionary in treating the even most aggressive brain cancer such as glioblastoma. They can prolong the duration of the drug retention in the systemic circulation [24]. During trials of nanomaterials (protein–polymer conjugates and liposomes), targeted cancer therapy started in the mid-1980s but was released to the market in the mid-1990s. Numerous nanomaterials for cancer therapy are under clinical trials [35,63,64,65,66,67,68].

### 4.2. Applicability of Nanobiotechnology in Brain Cancer

In the 1980s, the enhanced permeability and retention (EPR) effect showing nanoparticles’ aggregation in tumor cells was first observed [69], while NPs’ ability to cross BBB was observed in 1995 [70]. Leaky endothelium is commonly observed in blood vessels of tumors due to rapid and defective angiogenesis, failure of its normal barrier function, and permitting macromolecules to enter. NPs (approx. size 30 and 100 nm) passively enter through the leaky vasculature into tumor tissue when delivered intravenously, accumulating in the tumor site because of defective lymphatic-drainage, and deliver therapeutics to the cancer cells (EPR effect) [71]. Various nanomaterials, including metallic nanoparticles, micelles, liposomes, and dendrimers, were also investigated for their applicability in brain tumor targeting [24]. Figure 2 shows the schematic representation of various nanomaterials used for brain cancer therapy.

### 4.3. Nanoparticles

The affinity of NPs toward the acidic environment that characterizes tumor tissue is a targeting strategy with NPs to promote more efficacy over the detection and treatment of cancer without affecting normal cells [72]. Due to their size, nanomedicine can easily flow across small capillaries and obtain access to specific tissue tumor cells. Besides, once a therapeutic molecule is added, it can also find the extent of the healing effect inside the brain tumor. However, the major obstacle for any drug that is intended to be delivered to the brain is the BBB. Different strategies have been used to deliver drugs across the BBB such as chemical modification of drugs and osmotic interruption of the BBB [73]. Similarly, modifications of NPs are equally important for the targeted delivery towards the brain via crossing the BBB [74,75,76].

Indeed, with the support of nanomaterials, new erratic and diagnostic investigation concepts, innovative diagnostics, and tailored methods could be combined and succeed. In 9L glioma cells, polyacrylamide nanoparticles with a dye such as a coomassie blue, indocyanine green, or methylene blue have produced distinct color changes [77].

Nanoparticles can spread to the leaky tumor tissue vasculature through the EPR effect. This cancer attribute is distinguished by inadequate lymphatic drainage causing nanoparticles to accumulate even more than plasma concentrations. Effective GBM therapy could be obtained by meeting 3 key objectives, i.e., enhancing chemotherapy agents’ ability to cross BBB, infiltrate brain tissue that enters the cancerous tissues in an appropriate amount, preventing/minimizing side effects, and preserving the therapeutic influence of medications on tumor site [66]. Silver nanoparticles (AgNPs) at non-cytotoxic doses can induce genes related to DNA damage, incorrect cell cycle progression, and cell death. AgNPs have induced DNA damage and increased tumor cell apoptosis, including glioblastoma multiforme, leading to arrest of G2/M cell cycle phase [78]. In the Human GBM cell line- U-87 supplemented with AgNPs (40 μg/mL) inhibited its proliferation by triggering cell-mediated apoptosis through the activation of caspase-3 and caspase-9 [78]. Ag/AgCl-NPs effectively blocked GBM02 cell line growth compared to TMZ by representing 82% and 62% inhibition, respectively [79].

Gold nanoparticles (AuNPs) have some unique characteristics with a core, which is biocompatible and which can be prepared to various shapes depending upon the theranostic applications by altering its chemical precursor, i.e., hydrogen tetrachloroaurate. They also are shown to have an oxidative stress-induced cytotoxic effect in cancer cells. Additionally, they exhibit excellent optical properties, which makes them more suitable multifarious theragnostic therapeutics for clinical applications [80,81]. Moreover, they specifically recognize the receptor target site with the therapeutic molecule loaded by covalent conjugation or electrostatic interaction [82]. Compared to radiation therapy alone, a new concept comprising AuNPs for radiation therapy has facilitated long-term recovery with enhanced glioma treatment [83,84]. In contrast, AuNPs could be a viable tool for defining the intraoperative tumor margin, which has improved brain cancer therapy remarkably [85].

Solid lipid nanoparticles (SLNs) are the nanocarriers that are made of biocompatible lipids with the potential for delivery to the targeted site. They are nothing but dispersion consisting of lipid in water or aqueous solution with surfactants with the typical size of 10-100 nm. SLNs also possesses the perks of polymeric nanoparticles and liposomes with higher stability when exposed to the physiological environment. Usage of toxic solvents has also been avoided in the SLNs preparation, which makes it safer to deliver hydrophilic and hydrophobic drugs with unique advantages over macromolecules and peptides [86,87].

#### 4.3.1. Magnetic Nanoparticles

Magnetic nanoparticles (MNPs) are usually having magnetic properties that could be targeted to a specific location by applying an external magnetic field. They are one of the most investigated nanoparticulate systems. MNPs are synthesized from coprecipitation of iron (II), or iron (III) has a core which is made up of either Fe_3_O_4_ or γ-coated by polymers like PEG, dextran, polyvinyl alcohol (PVA), chitosan and polyvinylpyrrolidone (PVP) [88,89]. The polymeric coating enables them to evade the immune cells. The cancer cells in the hypoxic regions are generally susceptible; however, overcoming the damage to normal cells remain an issue and has toxicity similar to other metallic nanomaterials. It also can form free radicals because of the free Fe^2+^ ion damaging the DNA and other molecules [24].

#### 4.3.2. Polymeric Nanoparticles

Polymeric nanoparticles (PNPs) have several advantages for delivering drugs to the brain because of their higher encapsulation capacity, thereby preventing its excretion and metabolism, and delivering chemotherapeutic agents without altering the BBB [90]. Using non-toxic or biocompatible polymers that inhibit therapeutic agents’ release inside normal cells reduces overall systemic toxicity and evades the reticuloendothelial system [81,91,92]. Understanding the drug and polymers’ physicochemical properties can lead to efficient polymeric nanoparticles’ design for brain-targeted drug delivery [93]. Polymers including poly(glycolic acid) (PGA), poly(ε-caprolactone) (PCL), poly(lactic acid) (PLA), poly(butyl-cyanoacrylate) (PBCA) and poly(amino acids) were the most employed polymers for targeting the brain owing to their biocompatibility and nontoxic properties in comparison to the other polymers [94]. Two components were built into the biomimetic nanoformulations with a core arrangement of PCL and pluronic copolymer F68 for indocyanine green (ICG)-loaded polymeric nanoparticles that could work as a photothermal agent and an imaging agent as well. Camptothecin-loaded PLGA NPs have presented a ten-time upsurge in their brain drug delivery efficacy and have enhanced the survival of syngeneic GL261 murine tumors [95]. As 4T1 mammary breast and B16F10 mouse melanoma cells were used to derive the shells, brain metastatic tumor cell membranes replicate the source cells’ highly complex functionalities. BBB disruption is a viable method for increasing the flow of drugs across the brain’s circulatory system, which can be done by three methods: ultrasonic disruption, osmotic disruption, and magnetic disruption. However, opening the BBB through osmotic pressure may elicit disruption in the CNS’s normal function; thus, it may allow the accumulation of the toxic substances. An ultrasound technique could be applied to unlock the BBB, which reversibly augment the drug’s disposition across the BBB effectively. For instance, 10-enriched L-4- boronophenylalanine-fructose (BPA-f) could be used to enhance GBM therapy with ultrasound-focused microbubbles that may be administered to the targeted site [96,97].

Various chemotherapeutic drugs and molecules are encapsulated or adsorbed on the surface of polymeric NPs, which are excellent vehicles for targeted drug delivery [98]. Glycolic and lactic acid were the polymers’ degradation by-products of the Krebs cycle and they are removed as H_2_O and CO_2_ from the human body [99]. PNPs have certain benefits over other nanoformulations, including enhanced kinetics release, improved compatibility with other molecules, devoid of phospholipids oxidation problems, and exhibits enhanced shelf life [100,101,102]. In 1995, the 1st polymer nanoformulation was developed to transport the medications across BBB by formulating as dalargin-PBCA nanoparticles [70]. p80 coated dalargin-loaded PBCA NPs were used to improve their permeation into cells of the brain [70,103]. This NPs composition was even used to deliver other medicines, i.e., loperamide and doxorubicin, to the brain [104]. Higher accumulation of the PEG-PHDCA (poly(hexadecyl cyanoacrylate)) NPs in the brain was juxtaposed to the p80 containing formulation, that can be facilitated by passive diffusion macrophage intake [105]. In an in vitro BBB model, transferrin-PLGA nanoparticles were 20 times higher than uncoated PLGA nanoparticles, taken up by BBB via endocytosis [106]. Synthetic protein nanoparticles (SONO), which are synthesized from human serum albumin orchestrated with iRGD peptide, have significantly inhibited the STAT-3 activation, resulting in their downregulation and critical regulator GBM’s progression. It has also showed a long-term survival rate of about 87.5% with radiation therapy; and it even triggered the immune system memory for GBM [107]. By adding the glioma membrane proteins to the NP surface, indocyanine green (ICG) loaded liposome (BLIPO-ICG) NPs can actively target the glioma by bypassing the BBB with the immune system evading characteristics. Augmented accretion of the NPs was detected on application with NIR Laser, suppressing and inhibiting the tumor cells’ growth by 94.2% proving its biomimetic theranostic application for imaging and therapy [108]. ICG loaded polymeric nanoparticles with an exterior shell consisting of PCL and pluronic F68 were synthesized, which could be applied for therapeutic and diagnostic investigational purposes as an imaging agent and photothermal agent. It has also shown efficacy against U68MG cells with increased penetration across the BBB [109].

#### 4.3.3. Liposomes

Liposomes (LPs) are of submicron-sized pouches made of single or multiple lipid bilayers with a hydrophilic chamber that is used as a nanocarrier system to carry the therapeutic agents after their discovery in early 1960s. Due to their unique and attractive biological and physico-chemical properties, they have been the most preferred nanocarriers for drug delivery. Nonetheless, their biocompatible nature is an add up to their wider implications and preference. LPs can entrap both hydrophobic and hydrophilic molecules in their membrane and inside the vesicles, respectively, providing access to cell compartments for delivering the various entrapped molecules. Adsorption of DNA and RNA can also be facilitated by using cationic lipids for liposome preparation [110]. They are also considered as safe by the immune system and have the crucial advantage of being nonimmunogenic without eliciting any immune response [111]. Liposomes based on the mechanism are categorized as passive, active, and physiochemical-directed liposomes. Liposomes for passive targeting were prepared from PEG and phospholipids at 5%:10% ratio that can bypass RES, which were <10 nm in size and were focused towards the brain tumor through EPR. Based on the structural characteristics via surface modifications, specific receptor targets can be targeted with their respective ligands, whereas physiochemical targeting can be done according to the tumor’s microenvironment, such as magnetic field, pH, and temperature, etc. [62].

The lipid-based nanoparticulate system can bypass BBB with EPR, can also bypass presystemic metabolism and BBB obstacles are managed with improved permeability and residence time. Various lipid systems are solid lipid nanoparticles (SLNs), suspensions, emulsions, hybrid polymeric nanomaterials, niosomes, nanocapsules, lipid polymer hybrid nanoparticles, liposomes, and nanostructured lipid carriers (NLCs), to name a few [112]. The transport mechanism for BBB targeted liposomes was classified as transporter-mediated transcytosis (TMT), receptor-mediated endocytosis (RME), adsorption mediated transcytosis (AMT), in which the peripheral endothelium has endocytosis/transcytosis activity across brain endothelium. Still, the drug recognition in the liposomes may cause its extrusion [76,113,114]. The amounts of loaded drugs can be reduced with efficient, targeted delivery systems in line with improved in vivo safety.

Along with their efficacy, lipid solubility enhancing techniques have the probability of enhanced molecular transport across BBB in the case of large biomolecules [115]. The liposome layer could be further modified to enhance blood supply and brain-specific distribution by adding macromolecules, including polymers, polysaccharides, peptides, antibodies, or aptamers. Unfortunately, there is no successful liposomal formulation targeting the brain. Nonetheless, some liposomal medicines are in the final stage of approval for clinical use, and few are accepted for clinical trials [116,117,118,119,120,121,122,123,124].

Brain-focused distribution of low to medium molecular weight neuropeptides via prodrug concept attached to a lipid carrier is a potential idea being worked out. The neurometabolic drug design strategy was gradually added to the brain-analogous thyrotropin release hormone [125]. Efforts have been taken to develop liposomes with specific ligands as receptor-mediated endocytosis are promising where transferrin receptors (TfRs) are expressed in the microvascular endothelial glioma cells [126,127]. An excellent anticancer effect was shown against glioma by the dual targeting cell-penetrating peptide (CPP) conjugated with DOX liposomes and tf. When it has been loaded with paclitaxel, even for DOX also it has shown to have increased targeting and therapeutic efficacy causing hemolysis and embolization [128,129,130,131]. When DOX loaded, Tf-LPs has shown efficient uptake by the GL261 cells and U87 cells, where approximate drug release of 50% have indicated its potential as a probable formulation for treating glioma [132]. CPPs have shown good cellular uptake with various therapeutic agents loaded inside, including small molecules to proteins. Innovative strategies such as PEGylation of the liposomes with the tumor-specific Ab have been shown to increase their specificity and time duration in the systemic circulation. R8-RGD peptide is the most favored ligand when liposomes are functionalized; it has increased tumor cell penetration and inhibited their proliferation post loading with paclitaxel [133,134,135,136,137,138,139].

Among the various transporter-mediated transcytosis proteins, glucose transporters (GLUT), have become the focal point for BBB-targeted liposomes. Formulation designed using lecithin/cholesterol/p-aminophenyl-alpha-mannoside in the ratio of 7:2:1, *v*/*v*/*v* has bypassed BBB via GLUT mediated transcytosis in in vivo experimentation [140]. SF6-filled liposome microbubbles conjugated with peptide ligands to IL-4R MB-Lipo (Dox)-IL4RTP have increased uptake of the brain tumor U87MG cell lines with antiproliferative effect. Effective targeting of MB-Lipo (Dox)-IL4RTP was observed through ultrasound imaging and inhibition of signal transduction peptides such as Ser15)-p53, p53, and p21 regulating cell cycle and DNA damage [141]. Temozolomide (TMZ) loaded to cationic liposome (CL) formulations with its biomolecular corona (BC) layer that forms on CLs after exposure to human plasma (HP). TMZ-loaded CL−BC complexes were enriched in BC fingerprints (BCFs) (e.g., Apolipoproteins, Vitronectin, and vitamin K-dependent protein), which can extensively bind to receptors that are produced in excess at the BBB. It is also shown to have increased accumulation in human umbilical vein endothelial cells (HUVECs), that are used as BBB for in vitro models. It could also be able to deliver TMZ to human glioblastoma U-87 MG cell line and have shown anticancer effect [142]. Additionally, TMZ loaded hydrogenated soy phosphatidylcholine (HSPC) liposomes with 105.7 ± 3.9 nm and 78.25 ± 0.98% of particle size and encapsulation efficiency, respectively, have depicted excellent cytotoxic effects against U-87 MG cell line indicating efficiency as a potent therapeutic with respect to the pristine drug [143]. Modification of liposomes in addition to PEGylation along with mAb against tfR, glial fibrillary acidic proteins, and delivery of sodium borocaptate and 5-fluorouracil (5-FU) has shown best effects against high-grade brain cancers [144,145,146]. Altered formulations like p-aminophenyl-iii-D-mannopyranoside (MAN) and transferrin conjugate daunorubicin liposomes and trans-activating transcriptional peptide (TATP) altered liposomes have also been used for the treatment of brain tumors [147,148].

#### 4.3.4. Polymeric Micelles

Polymeric micelles (10 to 100 nm) have a similar architecture as NPs. They are used in cancer treatment when they tend to form micelle at a concentration higher than the essential micellar concentration in an aqueous solution, as they tend to inhibit the efflux of drugs and cross BBB. They have a core made up of hydrophobic polymers such as poly(propylene glycol) (PPG), poly(caprolactone) with their hydrophilic external shell [149]. Along with the benefits of being kinetically and thermodynamically stable with small size (<50 nm), various studies are in progress to study the targeting efficacies and clinical trials of polymeric micelles formulations [150,151,152]. A formulation containing PTX-containing cyclic (Arginine-Glycine-Aspartic acid-D-Tyrosine-Lysine)-polyethylene glycol-block poly lactic acid-paclitaxel micelles C(RGDyK)-PEGPLA-PTX have displayed a double-fold surge in the therapeutic efficacy of GBM treatment. C(RGDyK)-PEG-PLA mice are scattered in the U87MG glioma model to the intracranial tumor that effectively suppresses tumor progression in the test PTX preparation. Study outcomes suggest that they are a potential nano-theranostic strategy for treating over-expressed glioblastoma with integrin αvβ3 [153].

#### 4.3.5. Dendrimers

Dendrimers (DRM) are hyperbranched three-dimensional molecular structures that are made up of an inner initiator core with multiple inner and outer layers [154,155,156]. They are categorized as generation “G” built on the number of branch cycles and their size [157]. Dendrimers have monodispersity with well chemical structures compared to conventional polymer nano vehicles. The unique dendrimer structure allows the drug loading by covalent conjugation or electrostatic adsorption [158]. The dendrimers’ physicochemical characteristics depend on their structural components, viz., central core, type of macromolecules and nature of functional groups. The techniques employed for the dendrimer synthesis are divergent and convergent, that is construction from the inside central core to outer periphery and vice versa [159]. In previous years, the critical focus of pharmaceutical research has been on developing efficient and straightforward ways of achieving increased effectiveness of the delivered drugs by examining different forms of nanoparticle (NP), contributing to a clear understanding of NP uptake in the brain [160]. Dendrimers have shown tremendous potential for non-invasive therapy among these accomplishments [161]. As Donald A. Tomalia designed, developed and studied a new class of multibranched poly(amidoamine) (PAMAM) structures that were referred to as ‘dendrimers’ in 1984, these nanomaterials were greatly explored for therapeutic uses [162]. Its branches’ structure has advantages such as nano spherical structure, low viscosity than the linear polymer of equivalent weight, small polydispersity index, and higher surface functionalization [157]. Different types of dendrimers like polyamidoamine (PAMAM), poly(propylenemine) (PPI), triazine and poly-L-lysine (PLL) and their conjugates were studied for targeted delivery of therapeutic agents and gene for cancer treatment.

#### 4.3.6. Quantum Dots

Quantum dots (QD) semiconductor materials are about size range of 2–10 nm, which are known for their outstanding optical properties with broad excitation and small excitation-emission spectra, and they are highly photostable too [163,164]. QD finds application in cancer imaging. It can excite to different lights by monochromatic light. As they say, in the excited state for a prolonged period, distinguished from the other backgrounds [165]. To solubilize, surface modification of QD with thiol groups is preferred with encapsulation in a micelle or bounding to polysaccharides or amphiphilic molecules. PEG coating is essential for evading the immune system, which allows easy passage across the BBB and QD penetrates cells due to the EPR effect. QD is used for both theranostic and imaging purposes, but its application in medicine raises various safety risks. QD with cadmium can be extremely toxic, which can overcome by using substitution elements that are suitable for encapsulation and free from cadmium. The silicon core of size <4 nm has been used for both internal scanning diagnosis and delivery of therapeutic agents and proven to be nontoxic with specific optical properties [166,167].

Advantages of nanobiotechnology based drugs over the available brain cancer medications:

The advances in diagnosis procedures along with treatment have not reduced the mortality rate due to brain cancer. The reason for this can be attributed to the failure of cancer detection in the initial stages of the disease. These advancements include early detection and treatment regimens taken up more specifically by brain cancer cells and have decreased adverse effects. Current drugs are not necessarily effective in various cancers due to poor delivery at the target site [168]. The therapy’s effectiveness arises from conflicts with pharmacology or toxicity and developed drug resistance [169,170].

Nanomedicine has many advantages over traditional drug products that are desirable. First, designing using established preparation techniques having multimodal functional properties with precisely targeted drug delivery that is achieved by linking or conjugating or surface coating, with various ligands or moieties. The possibility to alter the size of nanocarrier for enhancing the entrapment of conjugated drugs or diagnostic molecules. Secondly, these systems are sufficiently capable of enhancing the existing drug’s pharmacokinetics and biodistribution features in the target tissues to increase their efficiency. Due to selective deposition in the cancerous cell with decreased delivery in healthy tissues, toxicity is also reduced due to unspecific biodistribution. Nanocarriers’ ability to deliver adequate amounts of loaded molecules specifically to cancer cells is critical for successful cancer detection and therapy [81,171,172,173]. In recent times, theranostic nanomedicines have been very promising, which can be exploited for detection and drug therapy at the molecular level to develop next-generation therapeutic agents [91,174,175]. Anti-GBM drugs loaded to the nanomaterial formulation can improve the conventional chemotherapeutic agents’ efficacy to treat various tumors. Some of the nano formulated drugs with improved efficacy are as follows: Nano formulations with the secondary metabolite product of algae have increased cytotoxicity towards A-71 GBM cells [176]. Liposome-mediated treatment of arsenic trioxide with the Mn has a good effect on TMZ resistant cancer cells in the brain [177]. PLGA copolymer conjugated with carboplatin resulted in higher tumor toxicity, resulting in a prolonged half-life in the porcine GBM rat model [178]. The efficacy of chlorotoxin increased without losing its therapeutic efficacy by loading into the iron oxide nanoparticles evading BBB and higher internalization by the cells [179]. The bioavailability and the water solubility of the curcumin enhanced in the dendrosome formulation with enhanced encapsulation, thereby suppressing the U87MG cell lines without affecting the normal cells [180]. Ciplastin loaded PEG-coated nanoparticles passed through the extracellular matrix with porous architecture between the GBM cells, attaining deeper penetration in tumor cells than ciplastin alone [181,182]. Doxorubicin (DOX) loaded polysorbate-coated nanoparticles were used to treat the rats affected with a 20% recurrence of GBM [181]. Another mode of brain tumor therapy is gene therapy, which used more RNA-associated nanoparticulate systems inducing apoptosis-induced cell death in Gbm and stoppage of cell movement by arresting the cell cycle regulation and cellular pathways [183,184,185,186].

### 4.4. Limitations of Nanobiotechnology Based Approaches

To date, liposomal preparations representing the first generation of nanomaterials have been used without any toxicity. The existence of other nanomedicines, such as phospholipids or biodegradable polymers, could also cause an adverse reaction that is still not clear [168]. In assessing potential harmful effects, the origin of the nanomaterial would probably a significant element. Although there is no definitive evidence of human toxicity to date, carbon nanotubes are expected to cause asbestos-like inflammation. Simultaneously, the nanomaterial is composed of heavy-metal that can reach and probably accumulate in vital organs, leading to specific tissue toxic effects [187]. While a solitary quantity of nanomaterials administering is under the heavy-metal poisoning limit, our failure to break it down can have effects that we cannot determine [188]. Metallic nanoparticles may create reactive oxygen species that make them poisonous and persist in the environment for long periods, with exposure leading to unexpected consequences. There are various threats to be considered based on the form of administration. The effect of nanomedicines on various blood components and their functioning needs to be checked for systematic administration, for instance [24].

While the implementation of nanomaterials in medicine seeks to improve the advantages of potential therapy for patients, their application poses concerns about threats they present to human patients and the environment [189]. It is still unclear if their main objective would be changed following interaction with biological fluids or molecules. As nanomaterials’ chemical and physical properties are not well known, their clinical studies’ presence should be closely controlled. Several techniques can define nanoparticles for safety reasons. This promotes creating a new specialized area of research called nanotoxicology, which can determine the possible toxicity of nanoparticles more suitably and sensitively. New assays will have to be designed to determine the structures and functioning of nanoparticles [66,187]. The physicochemical properties contributing to its therapeutic action interact with the biological and immunological system. Various scientific reports suggest that it can modulate immunity, antigenicity and even cause autoimmunity. Cytokine production through Th1/Th2 was highly influenced by various NPs in its pattern of production and even induce IgE production inducing allergic reactions. It also has been reported that various increase proinflammatory cytokines and chemokines such as IL-1β, MIP-1alpha, and GM-CSF, MIP-1α, respectively, influencing various signaling pathways [190]. It can take a lot of time to detect and understand the adverse effects in humans. Preferably, humans must be subjected to nanoparticles’ desired concentrations for a sufficiently prolonged time to assess toxicity effectively. Frequent administering and constant need for structurally diverse nanomaterials for such studies require more distilled medicines. When administering considerations such as biological complexity, disease heterogeneity, biological barriers, and aggregation in normal tissues, should be undertaken and nanomedicines should possess long shelf life and durability [68].

Various methods have been applied to target the BBB for the non-invasive delivery of the drugs. It has been reported that PLGA NPs functionalized nanoparticles with GALC targeted to Angiopep-2 (Ang2), g7, or transferrin binding (Tf2) peptides to bypass BBB and reinstated levels to their actual levels as in controls Twitcher mouse depicting the recovery of enzymatic activity from lysosomal storage disorder (LSD) in CNS environment [191]. In addition, the g7 polymeric NPs loaded with the iduronate 2-sulfatase (IDS), have attenuated the LSD condition in vitro and in vivo with a marked reduction in the levels of GAG proteins seems to an alternate therapeutic approach for the enzyme delivery across the BBB [192]. In GBM therapy, aptamers are shown to have a prominent application evident from various in vitro and in vivo experiments attributed to their ideal properties [193]. Biological therapy with infusing TXB2-hFc to neurotensin in mice does not induce any adverse effect or specific clearance from organs. It is also shown that TXB2 is a selective variable domain of new antigen receptor (VNAR) to transferrin receptor 1 (TfR1) crossing BBB with safe pharmacological and pharmacokinetic profile [194].

### 4.5. Can Nanobiotechnology Fulfill the Current Need for Alternate Brain Cancer Therapy?

Conventional strategies such as surgical resection, chemotherapy, and radiotherapy have limited application as per various reports due to the tumor’s metastasizing and complex nature. Due to the conventional treatments’ inefficiency, various new strategies were developed to improve the diagnosing and therapy, thereby minimizing toxicity and distributing the drugs to the normal tissues. In today’s scenario, nanotechnology-based therapeutics has emerged as the best choice for the conventional treatments of brain tumors with several advantages (Table 1).

In the future, it is expected that nanomaterials could surpass the current strategies for the treatment. It has also shown an increase in and regulates pharmacokinetics and drug distribution regulation with increased penetration across the BBB. Due to their precise surface area and architecture, they can assist a high payload of anticancer agents, ligands for targeting through conjugation or encapsulation. Nanobiotechnology seems to be a promising tool for brain cancer therapy over conventional treatment and imaging platforms. It has been shown to improve the efficacy, but on further investigation, optimization of their characteristic features, and with proper insights into the molecular mechanism governing the cancer pathogenesis, it will be an efficient method for treating brain cancer.

### 4.6. Prospects of Nanobiotechnology

While it has been shown to have considerable therapeutic efficacy, conventional treatment modalities do not stop the disease’s recurrence and its toxic side effects. Since the FDA has approved some therapeutic drugs and techniques, the overall survival rates have increased by months. Nanotechnology has made drug delivery to the brain possible. Some clinical studies have been presented in Table 2.

Brain drug delivery using nanotechnology has its own advantages and disadvantages. The amount and extent of drug delivery could be increased by drug lipidization. The formulation type is heavily dependent on the diseased state. The medications used for glioblastoma therapy are extremely cytotoxic. They must also be precisely aimed at the affected area. It is challenging to extract nanostructured drugs from the tissues, which can lead to drug accumulation and subsequent damage to the tissues. The nanocarriers below 100 nm alone can pass through the BBB. However, during the designing of these minute carriers, the anticipated loss of drug is high, which increases the overall cost. Carriers’ surface alteration often causes a fractional or complete loss of drug activity.

Differential targeting of cerebral regions must also resolve the side effects of non-specific binding. Nanomaterials’ stability and durability would be critical in targeted delivery, as non-interventional for any part of the brain are highly favored. 

There are scant studies on the clinical aspects of nanomaterials in drug delivery. Many of the formulations developed have been a success at laboratory scale, but await regulatory approval for clinical trials. As a consequence, therapeutic agents delivered to the brain via nanocarriers are in the preliminary phase and need comprehensive clinical evaluation for the pragmatic treatment of various brain cancers. In addition to the amalgamation of bionanotechnology with traditional tumor cell therapy, significant developments have been observed in tumor cells, focusing on the treatment of complex tumors. Polymeric nanoparticles attract further interest in treating metastatic gliomas due to their biocompatible and biodegradable behavior within the body and limitless models and characteristics that can be engineered.

## 5. Conclusions

On the basis of data analytics of the study population, it has been established that the gliomas, astrocytoma, meningioma, and metastatic cancer have a higher incidence rate in Saudi Arabia. The outcomes of numerous studies mentioned above reflect the prevalence and incidence demographics of brain tumor in the Saudi Arabia region, and thereby offers a distinctive platform to percept and propose the futuristic insights. Taking into account the incidence demographics of brain cancer among distinct populations, its complex nature, heterogeneity and invasiveness, it is evident that treating brain cancer is quite arduous. Nevertheless, the nanomaterials seem to be a promising approach that can overcome the shortcomings of the conventional therapies, wherein the polymer-based nanomaterials are conjugated with an array of nanomaterials to treat brain cancer and have shown promising results. Thus, for implication in efficient anti-cancer therapy, designing nanomaterials to be biocompatible and biodegradable is of crucial importance. Besides that, drug buildup in non-cancerous organs should be inhibited aiding high excretory rates to assure minimum damage to healthy tissues. To conclude, with the rigorous clinical investigation and absolute fixing of the toxic side effects, nanobiotechnology seems to hold a promise for successful brain cancer therapy in near future.

## 6. Recommendations

The nanotechnology-based therapeutic delivery systems can be used to effectively counter the drawbacks of the conventional treatment strategies. However, the success of foregoing drug delivery platforms depends on these nanomaterials being highly target-specific, biocompatible and bioresorbable. The clinical trials of these nano-based formulations for fixing the adverse effects need to be worked out. Drawing from on our study, we recommend the use of polymeric nanoparticles and liposomes based combinational therapy that could be positively exploited for effective treatment and management of brain cancer.

## Figures and Tables

**Figure 1 life-11-00421-f001:**
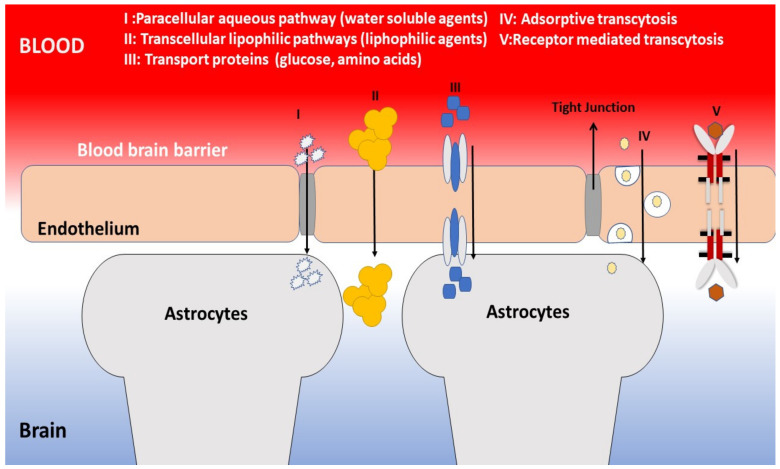
Pathways targeted by nanomaterials to travel across BBB.

**Figure 2 life-11-00421-f002:**
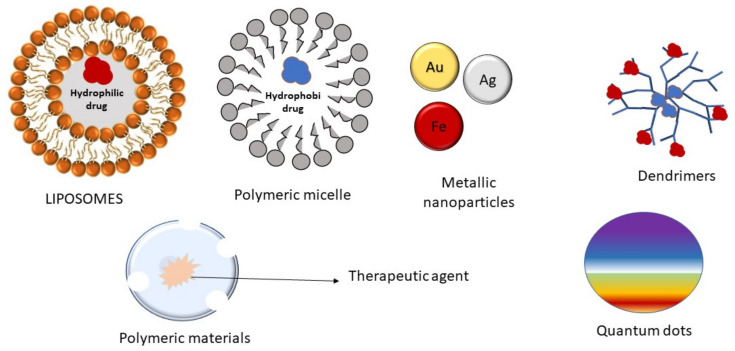
Schematic representation of different nanomaterials for brain cancer therapy: Polymers, micelles, liposomes, dendrimers, gold, silver, and iron nanoparticles, and quantum dots.

**Table 1 life-11-00421-t001:** Advantages of nanomaterials for brain cancer therapy.

Nanomaterials	Activity	Advantages	References
**Silver Nanoparticles**	Inhibition of glioblastoma multiforme cells proliferation with proapoptotic effect.	Antibacterial and antiviral properties	[72]
**Gold Nanoparticles**	Increased DNA damage in glioblastoma multiforme cells when combined with radiation therapy. Increases survival of brain tumor infected mice in comparison to radiation therapy alone.	Surface plasmon resonance Absorb light in the near-infrared region Easy surface modification Targeted drug delivery CT contrast agent	[24,77,78,195]
**Magnetic nanoparticles**	Thermotherapeutic effect on recurrent malignant brain tumors. Direct tumor targeting by peptide based surface modification.	Magnetically assisted drug therapy Conjugated drug therapy	[12,196]
**Quantum dots**	Photodynamic therapeutic effect on glioma cells. Bioimaging and diagnosis of glioma/ brain tumor.	Can bypass BBB Unique optical properties Bioconjugation and surface modifiable	[33,197]
**Liposomes**	Enhanced anti-glioma activity when co-modified with transferin. Targeted delivery to glioma cells achieved through peptide modification/conjugation.	Biocompatible Non-toxic and biodegradable Penetrate the BBB Physico-chemical targeting effects Surface modifiable	[62,126,129,133]
**Dendrimers**	Transcytosis based delivery to brain after bioconjugation with streptavidin adapter. Surface modification of dendrimers improved drug-release, compatibility and targeting towards brain tumor.	Monodispersity Very small size Enhances biocompatibility Controlled drug release Suitable for antibody and nucleic acid delivery Used in imaging	[153,155]
**Polymeric nanomaterials**	Ability to cross BBB without altering it. PLGA NPs presented a ten-time upsurge in their brain drug delivery efficacy and enhanced the survival of tumor infected murine.	Excellent nanocarriers in gene therapy Most extensively studied biodegradable and biocompatible nanocarriers to deliver brain tumor drugs The smaller size and higher structure solubility than liposomes Able to cross BBB Higher encapsulation efficiency Higher surface modification	[34,84,88,89]

**Table 2 life-11-00421-t002:** Clinical studies on injected nanomaterials for brain cancer.

Nanoparticle Type (Study Number)	Phase	Formulation	Disease Condition	References
**Polymer-gadolinium chelates (NCT02820454)**	II	AGuIX (polysiloxane gadolinium-chelates-based nanoparticles) concurrently with whole brain radiation	Brain metastases	[7,198]
**Gold (NCT03020017)**	Early phase-I	NU-0129	Glio sarcoma, recurrent glioblastoma	
**Cationic liposomes (NCT02340156)**	II	Liposomes encapsulated p53 cDNA in combination with oral temozolomide	Recurrent glioblastoma	
**Silica (NCT01266096)**	Micro dosing	124I-cRGDY-PEG-dots for PET scan	Recurrent metastatic melanoma, malignant brain tumor	
**Liposome (NCT00734682)**	I	CPT-11	CPT-1	

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
