# Peer review of "Current Status of Brain Tumor in the Kingdom of Saudi Arabia and Application of Nanobiotechnology for Its Treatment: A Comprehensive Review"

_life, 2021, doi:10.3390/life11050421_

Round 1

Reviewer 1 Report

Authors encompass the prevalence and treatment modalities available in the Kingdom of Saudi Arabia. Interestingly, they also provide a comprehensive analysis of the application of various nanotechnology-based products for brain cancer treatments and their future perspectives.

They appropriately identified and summarize current status of brain cancer in Saudi Arabia and scope of nanobiotechnology in its treatment.

They interestingly found that depending upon the study population, gliomas, astrocytoma, meningioma, and metastatic cancer have a higher incidence rate in Saudi Arabia, which follows the conventional treatment modalities for the brain cancer.

They appropriately conclude that nanomaterials possess the capability of surpassing the shortcomings of the conventional therapies. Moreover, some points that are listed below could be better discussed for the final publication in the journal Life, which in my opinion is appropriate for the current review article.

- lane 42= Of which…. difficult to read. Reformulate better.

- lane 113= what means “iatric benefits”?

- fig. 1 = words at the top of the picture have poor resolution

- lane 316 and 318 = references is needed

- Paragraph 4.3 = authors not discuss appropriately the main obstacle for the brain delivery of the nanoparticles : the presence of the BBB. Untargeted nanoparticles are not able to pass the BBB.

Authors have to maybe add a paragraph where they list the non invasive approach currently used to target the BBB: targeting nanoparticles with peptides as g7, ang2 or Tf2 (Del Grosso, Ambra, et al. "Brain-targeted enzyme-loaded nanoparticles: A breach through the blood-brain barrier for enzyme replacement therapy in Krabbe disease." Science advances 5.11 (2019): eaax7462. ; Rigon, L., Salvalaio, M., Pederzoli, F., Legnini, E., Duskey, J. T., D’Avanzo, F., ... & Tomanin, R. (2019). Targeting brain disease in MPSII: preclinical evaluation of IDS-loaded PLGA nanoparticles. International journal of molecular sciences20(8), 2014.) , aptamers (Cesarini, V., Scopa, C., Silvestris, D. A., Scafidi, A., Petrera, V., Del Baldo, G., & Gallo, A. (2020). Aptamer-Based In Vivo Therapeutic Targeting of Glioblastoma. Molecules25(18), 4267.) or antibodies (Stocki, P., Szary, J., Rasmussen, C. L., Demydchuk, M., Northall, L., Logan, D. B., ... & Rutkowski, J. L. (2021). Blood‐brain barrier transport using a high affinity, brain‐selective VNAR antibody targeting transferrin receptor 1. The FASEB Journal35(2), e21172.).

- Paragraph 4.4 = authors may add the point of the possible immunotoxicity reaction to nanoparticles administration (Di Gioacchino, M., Petrarca, C., Lazzarin, F., Di Giampaolo, L., Sabbioni, E., Boscolo, P., ... & Bernardini, G. (2011). Immunotoxicity of nanoparticles. International journal of immunopathology and pharmacology24(1 Suppl), 65S-71S.)

- Authors in general may indicates material which are already approved by FDA for nanoparticles formulations, as PLGA.

Author Response

Dear Honorable Reviewer,

We have revised the manuscript in the light of your valuable suggestions and  submitted the revised version with response letter for your kind perusal.

We take the opportunity to thank the honorable reviewer for the positive criticisms and suggestions, which have undoubtedly improved the manuscript.

*Note: All the changes are highlighted in yellow in the revised MS.

Reviewer 1:

Comment 1: - lane 42= Of which…. difficult to read. Reformulate better.

Rebuttal 1: As per the suggestion of the honorable reviewer, the sentence has been reformulated

Comment 2: - lane 113= what means “iatric benefits”?

Rebuttal 2: iatric benefit has been modified to therapeutic benefit to make it more clear. However, the word iatric means medical or related to medicine.

Comment 3: - fig. 1 = words at the top of the picture have poor resolution

Rebuttal 3: As per the suggestion of the honorable reviewer, the figure 1 has been duly modified.

Comment 4:  - lane 316 and 318 = references is needed.

Rebuttal 4: As per the suggestion of the honorable reviewer, required references were added as follow:

[69] Matsumura, Y.; Maeda, H. A new concept for macromolecular therapeutics in cancer chemotherapy: mechanism of tumoritropic accumulation of proteins and the antitumor agent smancs. Cancer Res. 1986, 46, 6387-92.

[70] Kreuter, J.; Alyautdin, R.N.; Kharkevich, D.A.; Ivanov, A.A. Passage of peptides through the blood-brain barrier with colloidal polymer particles (nanoparticles). Brain Res. 1995, 674(1), 171-4, doi: 10.1016/0006-8993(95)00023-j.

Comment 5: - Paragraph 4.3 = authors not discuss appropriately the main obstacle for the brain delivery of the nanoparticles: the presence of the BBB. Untargeted nanoparticles are not able to pass the BBB.

Rebuttal 5: As per the suggestion of the honorable reviewer, BBB aspects have been discussed and following references have been duly incorporated in the MS.

[73] Zhou, Y.; Peng, Z.; Seven, E.S.; Leblanc, R.M. Crossing the blood-brain barrier with nanoparticles. J Control Release 2018, 270, 290-303, doi: 10.1016/j.jconrel.2017.12.015.

[74] Saraiva, C.; Praça, C.; Ferreira, R.; Santos, T.; Ferreira, L.; Bernardino, L. Nanoparticle-mediated brain drug delivery: Overcoming blood-brain barrier to treat neurodegenerative diseases. J Control Release 2016, 235, 34-47. doi: 10.1016/j.jconrel.2016.05.044.

[75] Gao, H. Progress and perspectives on targeting nanoparticles for brain drug delivery. Acta Pharmaceutica Sinica B 2016, 6(4), 268-286.

[76] Zhang, T.T.; Li, W.; Meng, G.; Wang, P.; Liao, W. Strategies for transporting nanoparticles across the blood-brain barrier. Biomater Sci. 2016, 4(2), 219-29. doi: 10.1039/c5bm00383k.

Comment 6: Authors have to maybe add a paragraph where they list the non invasive approach currently used to target the BBB: targeting nanoparticles with peptides as g7, ang2 or Tf2 (Del Grosso, Ambra, et al. "Brain-targeted enzyme-loaded nanoparticles: A breach through the blood-brain barrier for enzyme replacement therapy in Krabbe disease." Science advances 5.11 (2019): eaax7462. ; Rigon, L., Salvalaio, M., Pederzoli, F., Legnini, E., Duskey, J. T., D’Avanzo, F., ... & Tomanin, R. (2019). Targeting brain disease in MPSII: preclinical evaluation of IDS-loaded PLGA nanoparticles. International journal of molecular sciences20(8), 2014.) , aptamers (Cesarini, V., Scopa, C., Silvestris, D. A., Scafidi, A., Petrera, V., Del Baldo, G., & Gallo, A. (2020). Aptamer-Based In Vivo Therapeutic Targeting of Glioblastoma. Molecules25(18), 4267.) or antibodies (Stocki, P., Szary, J., Rasmussen, C. L., Demydchuk, M., Northall, L., Logan, D. B., ... & Rutkowski, J. L. (2021). Blood‐brain barrier transport using a high affinity, brain‐selective VNAR antibody targeting transferrin receptor 1. The FASEB Journal35(2), e21172.).

Rebuttal 6: As per the suggestion of the honorable reviewer, non invasive approaches applied for crossing BBB has been added  and references has been included in the text.

Comment 7: - Paragraph 4.4 = authors may add the point of the possible immunotoxicity reaction to nanoparticles administration (Di Gioacchino, M., Petrarca, C., Lazzarin, F., Di Giampaolo, L., Sabbioni, E., Boscolo, P., ... & Bernardini, G. (2011). Immunotoxicity of nanoparticles. International journal of immunopathology and pharmacology24(1 Suppl), 65S-71S.)

Rebuttal 7: As per the suggestion of the honorable reviewer, possible immunotoxicity reaction to nanoparticles administration has been added and reference has been included.

Comment 8: - Authors in general may indicates material which are already approved by FDA for nanoparticles formulations, as PLGA.

Rebuttal 8: As per the suggestion of the honorable reviewer, commonly used polymers used for preparing nanoparticles for brain targeting was mentioned and highlighted in the manuscript.

Reviewer 2 Report

This is an interesting review about the current status of brain cancers in Saudi Arabia. The topic of the manuscript is of great interest for scientists working in and interdisciplinary context, including chemists, physicists, biologists, and oncologists.

Nevertheless, some improvements are required before publication. This reviewer is suggesting authors to consider the following comments for revising the paper:

  1. A summarizing table should be inserted to highlight not only the advantages of nanomaterials for brain cancer therapy (well collected in table 1), but also the most relevant examples available in the literature and cited in the text. This can help readers to have an overview of the state of the art in the field.
  2. Authors should clarify why they did not consider other kind of nanoparticles, such as carbon or zinc nanoparticles

Author Response

Dear Honorable Reviewer,

We have revised the manuscript in the light of your valuable suggestions and  submitted the revised version with response letter for your kind perusal.

We take the opportunity to thank the honorable reviewer for the positive criticisms and suggestions, which have undoubtedly improved the manuscript.

*Note: All the changes are highlighted in yellow in the revised MS.

Reviewer 2:

Comment 1: A summarizing table should be inserted to highlight not only the advantages of nanomaterials for brain cancer therapy (well collected in table 1), but also the most relevant examples available in the literature and cited in the text. This can help readers to have an overview of the state of the art in the field.

Rebuttal 1: As per the suggestion of the honorable reviewer, the table 1 of the review has been duly modified by adding a column of activity with reference.

Comment 2: Authors should clarify why they did not consider other kind of nanoparticles, such as carbon or zinc nanoparticles.

Rebuttal 2: We appreciate the concern of honorable reviewer. Most humbly, we would like to state that neurotoxicity of certain nanoparticles is well established. Thus, we avoided the nanoparticles which have lesser feasibility to be transformed into future nanomedicines for gliomas, based on the available literatures. With due respect, the following references are listed for your kind perusal.

  • da Rocha AM, Kist LW, Almeida EA, Silva DGH, Bonan CD, Altenhofen S, Kaufmann CG Jr, Bogo MR, Barros DM, Oliveira S, Geraldo V, Lacerda RG, Ferlauto AS, Ladeira LO, Monserrat JM. Neurotoxicity in zebrafish exposed to carbon nanotubes: Effects on neurotransmitters levels and antioxidant system. Comp Biochem Physiol C Toxicol Pharmacol. 2019 Apr;218:30-35. doi: 10.1016/j.cbpc.2018.12.008. Epub 2018 Dec 11. PMID: 30543862.
  • Gholamine B, Karimi I, Salimi A, Mazdarani P, Becker LA. Neurobehavioral toxicity of carbon nanotubes in mice. Toxicol Ind Health. 2017 Apr;33(4):340-350. doi: 10.1177/0748233716644381. Epub 2016 Jul 9. PMID: 27230352.
  • Bussy C, Al-Jamal KT, Boczkowski J, Lanone S, Prato M, Bianco A, Kostarelos K. Microglia Determine Brain Region-Specific Neurotoxic Responses to Chemically Functionalized Carbon Nanotubes. ACS Nano. 2015 Aug 25;9(8):7815-30. doi: 10.1021/acsnano.5b02358. Epub 2015 Jun 26. PMID: 26043308.
  • Tian L, Lin B, Wu L, Li K, Liu H, Yan J, Liu X, Xi Z. Neurotoxicity induced by zinc oxide nanoparticles: age-related differences and interaction. Sci Rep. 2015 Nov 3;5:16117. doi: 10.1038/srep16117. PMID: 26527454; PMCID: PMC4630782.
  • Yaqub, A., Faheem, I., Anjum, K.M. et al. Neurotoxicity of ZnO nanoparticles and associated motor function deficits in mice. Appl Nanosci 10, 177–185 (2020). https://doi.org/10.1007/s13204-019-01093-3
  • Amara S., Ben-Slama I., Mrad I., Rihane N., Jeljeli M., El-Mir L., Ben- Rhouma K., Rachidi W., Sčve M., Abdelmelek H., Sakly M., Acute exposure to zinc oxide nanoparticles does not affect the cognitive capacity and neurotransmitters levels in adult rats, Nanotoxicology, 2014, 8 Suppl 1, 208-15.
  • Amara S., Slama I.B., Omri K., El Ghoul J., El Mir L., Rhouma K.B., Abdelmelek H., Sakly M., Effects of nanoparticle zinc oxide on emotional behavior and trace elements homeostasis in rat brain, Toxicol. Ind. Health, 2015, 31(12), 1202-9.
  • Teleanu DM, Chircov C, Grumezescu AM, Teleanu RI. Neurotoxicity of Nanomaterials: An Up-to-Date Overview. Nanomaterials (Basel). 2019;9(1):96. Published 2019 Jan 13. doi:10.3390/nano9010096

Round 2

Reviewer 2 Report

Authors well answered to all comments in a proper manner. I recommend publication of the paper in its current form